# Feasibility of CPAP application and variables related to worsening of respiratory failure in pregnant women with SARS-CoV-2 pneumonia: Experience of a tertiary care centre

**Paola Faverio**[1‡]*, **Sara Ornaghi**[2‡], **Anna Stainer**[1], **Francesca Invernizzi**[2], **Mara Borelli**[1], **Federica Brunetti**[3], **Laura La Milia**[2], **Valentina Paolini**[1], **Roberto Rona**[1,4], **Giuseppe Foti**[1,4], **Fabrizio Luppi**[1], **Patrizia Vergani**[2], **Alberto Pesci**[1]

1 Respiratory Unit, School of Medicine and Surgery, University of Milano Bicocca, San Gerardo Hospital, ASST Monza, Monza, Italy, 2 Obstetric Unit, School of Medicine and Surgery, University of Milano Bicocca, MBBM Foundation Onlus at San Gerardo Hospital, Monza, Italy, 3 Department of Obstetrics and Gynaecology, Desio Hospital, ASST Monza, Desio, Italy, 4 Department of Anesthesia and Intensive Care Medicine, ASST Monza, Monza, Italy

‡ PF and SO are joint first authors on this work.
* paola.faverio@unimib.it

## Abstract

Continuous positive airway pressure (CPAP) has been successfully applied to patients with COVID-19 to prevent endotracheal intubation. However, experience of CPAP application in pregnant women with acute respiratory failure (ARF) due to SARS-CoV-2 pneumonia is scarce. This study aimed to describe the natural history and outcome of ARF in a cohort of pregnant women with SARS-CoV-2 pneumonia, focusing on the feasibility of helmet CPAP (h-CPAP) application and the variables related to ARF worsening. A retrospective, observational study enrolling 41 consecutive pregnant women hospitalised for SARS-CoV-2 pneumonia in a tertiary care center between March 2020 and March 2021. h-CPAP was applied if arterial partial pressure of oxygen to fraction of inspired oxygen ratio ($PaO_2/FiO_2$) was inferior to 200 and/or patients had respiratory distress despite adequate oxygen supplementation. Characteristics of patients requiring h-CPAP *vs* those in room air or oxygen only were compared. Twenty-seven (66%) patients showed hypoxemic ARF requiring oxygen supplementation and h-CPAP was needed in 10 cases (24%). $PaO_2/FiO_2$ was significantly improved during h-CPAP application. The device was well-tolerated in all cases with no adverse events. Higher serum C reactive protein and more extensive ($\geq$3 lobes) involvement at chest X-ray upon admission were observed in the h-CPAP group. Assessment of temporal distribution of cases showed a substantially increased rate of CPAP requirement during the third pandemic wave (January-March 2021). In conclusion, h-CPAP was feasible, safe, well-tolerated and improved oxygenation in pregnant women with moderate-to-severe ARF due to SARS-CoV-2 pneumonia. Moderate-to-severe ARF was more frequently observed during the third pandemic wave.

**Data Availability Statement:** All relevant data are within the manuscript and its Supporting Information files.

**Funding:** The authors received no specific funding for this work.

**Competing interests:** The authors have declared that no competing interests exist.

**Abbreviations:** ABG, arterial blood gas; ARF, acute respiratory failure; ARDS, acute respiratory distress syndrome; COVID-19, coronavirus disease 2019; CPAP, continuous positive airway pressure; CRP, C-reactive protein; CS, caesarean section; CT, computed tomography; ETI, endotracheal intubation; $FiO_2$, fraction of inspired oxygen; $HCO_3$, bicarbonates; h-CPAP, helmet continuous positive airway pressure; IMV, invasive mechanical ventilation; $PaCO_2$, arterial partial pressure of carbon dioxide; $PaO_2/FiO_2$, arterial partial pressure of oxygen to fraction of inspired oxygen ratio; PEEP, positive end expiratory pressure; RT-PCR, Real Time Polymerase Chain Reaction; $SaO_2$, arterial oxygen saturation of haemoglobin; SARS-CoV-2, severe coronavirus 2.

## Introduction

Severe acute respiratory syndrome coronavirus 2 (SARS-CoV-2) infection causing coronavirus disease 2019 (COVID-19) was declared a global pandemic in March 2020. COVID-19-related pulmonary manifestations are broad, ranging from mild respiratory symptoms without supplemental oxygen requirements to pneumonia and acute respiratory distress syndrome (ARDS) with severe acute respiratory failure (ARF) [1]. In more severe cases, the use of continuous positive airway pressure (CPAP) through helmet and prone positioning has been largely described [2, 3].

Helmet CPAP (h-CPAP) has been recommended in recent guidelines as the first non-invasive respiratory support choice for patients with COVID-19 and moderate-to-severe ARF [4, 5]. Its success is mainly due to the possibility of being applied outside the intensive care unit and to prevent the need for endotracheal intubation (ETI). Furthermore, helmet has the least amount of particle dispersion and air contamination among all noninvasive devices.

Direct effects of SARS-CoV-2 infection on pregnant women and their newborns have been extensively studied, with findings unanimously suggesting that moderate to severe symptomatic infection associates to adverse obstetric outcomes, including preterm birth <37 weeks and low birthweight [6, 7]. Medical intervention due to worsening of maternal ARF and need of ETI has been identified as the main risk factor for these complications [8].

In this context, the application of h-CPAP with the aim to non-invasively manage moderate-to-severe ARF, thus possibly avoiding iatrogenic preterm birth, would be of substantial interest. However, the experience regarding this approach is limited to only a few case reports [9, 10]. In the available literature, evidence supports the feasibility of h-CPAP application in adult patients with community acquired pneumonia [11]. In contrast, no evidence is available regarding helmet CPAP utilization and feasibility in pregnant women.

Here we describe the natural history and outcome of respiratory failure in a cohort of consecutive pregnant women hospitalised for SARS-CoV-2 pneumonia in a tertiary care center, with a particular emphasis on the feasibility of h-CPAP application and possible variables related to ARF worsening.

## Material and methods

This was a retrospective study on consecutive pregnant women admitted to the MBBM foundation / San Gerardo University hospital, a tertiary care centre in Monza, Italy, with a diagnosis of SARS-CoV-2 pneumonia between March 1st, 2020 and March 31st, 2021. Women accessing our Emergency Department and identified as SARS-CoV-2 infected but with no evidence of pneumonia were excluded.

According to our Institutional protocol, all women with SARS-CoV-2 infection identified by Real Time Polymerase Chain Reaction (RT-PCR) on nasopharyngeal swab were required to perform a chest X-ray, as well as a complete clinical and laboratory evaluation.

Oxygen supplementation was started if arterial oxygen saturation of haemoglobin ($SaO_2$) was <95% or if respiratory rate was ≥20 breaths/minute either at rest or after a six-minute walking test. In addition, thromboprophylaxis by low molecular weight heparin was started in all patients, and dosed according to maternal weight (4000 IU if <90 Kg, 6000 IU if ≥90 Kg, once daily).

h-CPAP was applied according to our Institutional protocol by the physician in charge if arterial partial pressure of oxygen to fraction of inspired oxygen ratio ($PaO_2/FIO_2$) was inferior to 200 and/or if patients had respiratory distress, including increased respiratory rate and/or thoracic-abdominal dyssynchrony, despite adequate oxygen supplementation. Positive end expiratory pressure (PEEP) responsiveness and titration was assessed through a h-CPAP trial

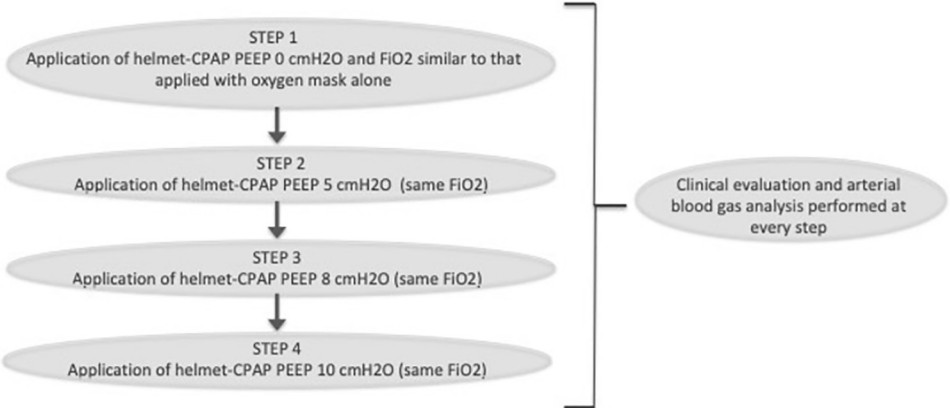

STEP 1
Application of helmet-CPAP PEEP 0 cmH2O and FiO2 similar to that applied with oxygen mask alone

STEP 2
Application of helmet-CPAP PEEP 5 cmH2O (same FiO2)

STEP 3
Application of helmet-CPAP PEEP 8 cmH2O (same FiO2)

STEP 4
Application of helmet-CPAP PEEP 10 cmH2O (same FiO2)

Clinical evaluation and arterial blood gas analysis performed at every step

**Fig 1. Description of the CPAP-trial used to differentiate PEEP responder patients from PEEP-non responder.**
Footnotes: PEEP = Positive End-Expiratory Pressure. CPAP = continuous positive airway pressure. FIO$_2$ = fractional concentration of oxygen in inspired air.

performed as explained by *Paolini et al.* [12], Fig 1. In brief, PEEP responsiveness was evaluated comparing clinical and arterial blood gas (ABG) parameters during oxygen supplementation with h-CPAP with PEEP 0 cm H$_2$O and other PEEP levels (5, 8 and 10 cm H$_2$O, respectively), maintaining the same fractional concentration of oxygen in inspired air (FiO$_2$). A PEEP-responder is defined as a subject with clinical and/or arterial blood gases improvement with h-CPAP PEEP 5, 8 or 10 cm H$_2$O compared to h-CPAP PEEP 0 cm H$_2$O, maintaining the same FIO$_2$. Pregnant patients requiring h-CPAP were transferred to the high-dependency respiratory unit and underwent continuous non-invasive monitoring.

During the first months of the pandemic (March–July 2020), SARS-CoV-2 infected women with pneumonia received hydroxychloroquine (200 mg twice daily for seven days), alongside a third-generation cephalosporin (2 gr, once daily for seven days) as antibiotic prophylaxis. Corticosteroid therapy (betamethasone 12 mg IM 24 hour apart) was administered only to induce fetal lung maturation in case of prematurity <34 weeks' gestation and imminent delivery.

During the subsequent months of the pandemic (August 2020 –March 2021) and after publication of the RECOVERY trial's preliminary data [13], corticosteroid therapy was administered to all women with SARS-CoV-2 pneumonia requiring oxygen supplementation (on admission start dosage was methylprednisolone 40 mg IV once daily for ten days, followed by oral therapy titration; however, in case of worsening of ARF requiring a h-CPAP trial and in the absence of bacterial infection, methylprednisolone was increased up to 1 mg/kg/die). In addition, use of hydroxychloroquine was discontinued from May 2020, due to the lack of efficacy in COVID-19 patients [14] and antibiotic therapy was started only if a superimposed bacterial infection was suspected.

Fetal monitoring in women with SARS-CoV-2 pneumonia was performed by non-stress-test (twice daily) and biophysical profile with ultrasound scan (once daily).

Information on patients' demographic characteristics, obstetric history, course of pregnancy, laboratory and radiological data, and respiratory parameters were collected by reviewing the electronic medical records (S.O., P.F., F.I., A.S. and M.B.) and registered in a dedicated logbook. Radiological involvement and extension at chest X-ray were reviewed independently by two pulmonologists and disagreements were resolved by a third senior pulmonologist.

The study was approved by the Institutional Review Board of the University of Milano Bicocca (3140/2020). Written informed consent was obtained by patients included in the study.

### Statistical analyses

Assessment of normality was performed by means of Kolmogorov–Smirnov test. Statistical analyses included Chi-Square test or Fischer Exact test for categorical variables, and independent Student's T-test or Mann-Whitney's test for continuous variables.

Statistical significance was set at $P < .05$ (SPSS software, version 27; Prism software, version 7).

## Results

### Clinical features of the study cohort

During the study period, 41 consecutive pregnant women were diagnosed with SARS-CoV-2 pneumonia and admitted to the MBBM Foundation / San Gerardo Hospital, Monza, Italy. Patients were stratified according to the maximum oxygen and ventilatory support required, Fig 2.

General and obstetric characteristics of the cohort are summarised in Table 1. All pregnancies were spontaneously conceived and none of the women were active smokers during gestation. In 5 (12.2%) cases, relevant comorbidities were identified, including diabetes mellitus (n = 1), asthma (n = 2), and chronic hypertension (n = 2). Overall, 21 (51.2%) women had at least one pregnancy-related complication diagnosed before hospital admission. Mean gestational age at hospital admission was 30.4 ± 5.4 weeks.

Four (9.8%) women were completely asymptomatic upon hospital admission, despite radiological evidence of pneumonia at chest X-ray, and they were referred to our Emergency Department after diagnosis of SARS-CoV-2 infection by nasopharyngeal swab performed because of a high-risk contact. All four patients developed symptoms of infection, including

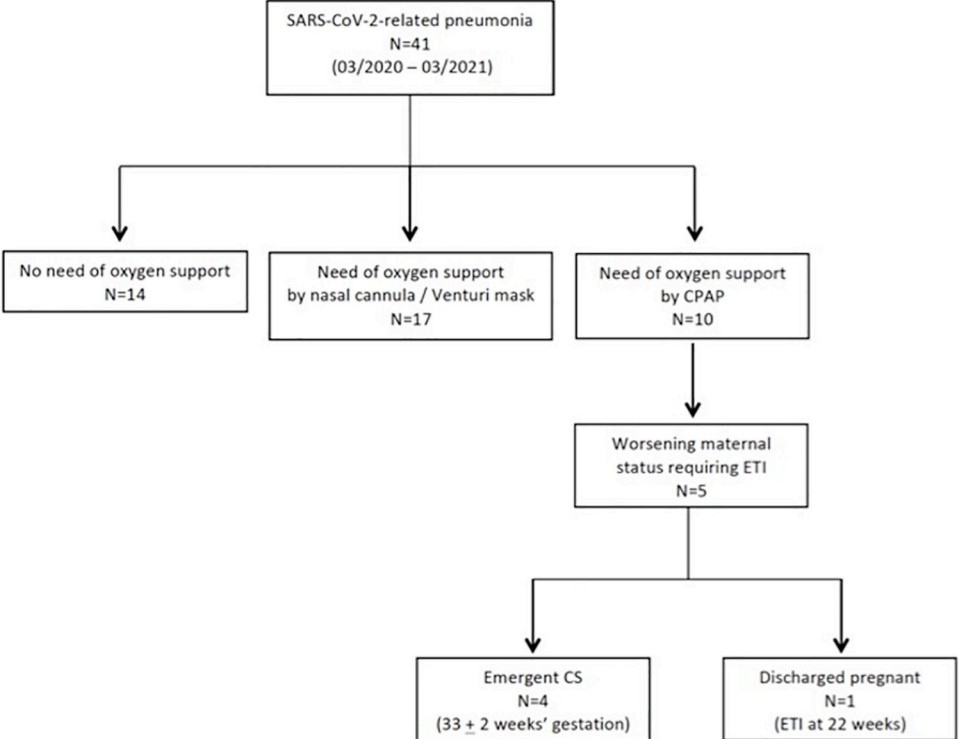

**Fig 2. Study flow chart.** Footnotes: CPAP = continuous positive airway pressure. ETI = endotracheal intubation. CS = caesarean section.

**Table 1. General characteristics of pregnant women with SARS-CoV-2-induced pneumonia managed at our university center.**

| Variables | Study population |
|---|---|
| | N = 41 |
| Maternal age (years) | 32.1 ± 5.3 |
| BAME | 10 (24.4) |
| Pregestational BMI (Kg/m$^2$) | 25.5 ± 5.0 |
| - ≥30 | 8 (19.5) |
| Nulliparity | 14 (34.1) |
| GDM | 13 (31.7) |
| HDP | 1 (2.4) |
| Cholestasis | 3 (7.3) |
| Threatening PTL | 5 (12.2) |

Footnotes: Data shown as mean ± standard deviation or number (percentage).

BAME, Black, Asian, and Minor Ethnicity; BMI, Body Mass Index; GA, gestational age; GDM, gestational diabetes mellitus; HDP, hypertensive disorders of pregnancy, including gestational hypertension and preeclampsia; PTL, preterm labor.

fever and cough, during the hospital stay. Among the remaining 37 patients, fever and cough were the most common symptoms reported at hospital admission (n = 26 each, 70.3%), followed by shortness of breath (n = 13, 35.1%), anosmia and ageusia (n = 8, 21.6%) and myalgia (n = 8, 21.6%). Overall, a high-risk contact was identified in 14 (34.1%) cases.

All patients underwent chest X-ray on hospital admission showing pulmonary infiltrates suggestive of pneumonia. Twenty-four (59%) cases had bilateral involvement, while lower lobes were more affected than mid-upper lobes: right and left lower lobes were affected in 28 and 23 cases, respectively, while right middle lobe was involved in 16 cases, lingula in 17 cases, right upper lobe in 4 cases and left upper lobe in 7 cases. Eight patients also performed chest computed tomography (CT) scan with contrast medium to rule out pulmonary embolism, which was diagnosed only in one case.

## Continuous positive airway pressure application

Twenty-seven (66%) patients showed hypoxemic ARF requiring oxygen supplementation and 10 of them subsequently required CPAP application, using helmet, because of deterioration of gas exchange and/or respiratory distress. ABG analyses on admission (T0), on oxygen therapy immediately before h-CPAP application (T1) and during first h-CPAP application (T2) (at best PEEP value identified during CPAP trial) are summarised in Table 2 and Fig 3. PaO$_2$, PaO$_2$/FIO$_2$, and arterial oxygen saturation of haemoglobin (SaO$_2$) were significantly improved during CPAP application, while no significant changes were observed in carbon dioxide arterial partial pressure (PaCO$_2$), serum bicarbonates (HCO$_3$), and lactates values.

After applying the h-CPAP trial, 8 out of 10 patients were identified as PEEP-responders. In these 8 patients h-CPAP was well tolerated and applied for three cycles per day (morning, afternoon and night), with a median (IQR) of 5 (4–9.3) days of h-CPAP use. However, three of these women showed clinical and ABG deterioration notwithstanding the h-CPAP, thus ultimately requiring ETI and invasive mechanical ventilation (IMV) on day 3, 4 and 10 after h-CPAP application, respectively. Best PEEP levels identified at h-CPAP trial were 10 cm H$_2$O in 4 cases, 7.5 cm H$_2$O in 1 case and 5 cm H$_2$O in 3 cases. The two out of ten patients identified as PEEP non-responders at the h-CPAP trial continued high flow oxygen administration but eventually required ETI and IMV.

**Table 2. Arterial blood gas results of the 10 patients who required continuous positive airway pressure application.**

| | on admission (T0) | pre-CPAP (T1) | best during CPAP trial (T2) | p (T1 vs T2) |
|---|---|---|---|---|
| FIO$_2$% | 21 (21–21) | 65 (45–70) | 50 (50–60) | |
| pH | 7.45 (7.41–7.47) | 7.44 (7.41–7.49) | 7.44 (7.42–7.45) | 0.89 |
| PaO$_2$ mm Hg | 80 (77.25–85.75) | 103.5 (75.5–131) | 175 (144–242.25) | 0.033 |
| PaO$_2$/FiO$_2$ ratio | 375.5 (366–388.5) | 153 (135–256.5) | 370 (283.5–425) | < 0.001 |
| PaCO$_2$ mm Hg | 27 (23.25–28) | 28.75 (26–30.25) | 28 (25.75–29.63) | 0.61 |
| HCO$_3$ mmol/L | 20.5 (15.75–21.25) | 21 (19.5–22.5) | 21 (19.25–22) | 0.38 |
| Lactates mmol/L | 1.1 (0.9–1.11) | 1.1 (0.8–1.75) | 0.9 (0.8–1.10) | 0.066 |
| SaO$_2$% | 96 (95.75–97.4) | 98 (95.5–98.5) | 99 (99–99.5) | 0.038 |

Footnotes: Data are expressed as median (IQR). FIO$_2$ = fractional concentration of oxygen in inspired air. PaCO$_2$ = arterial partial pressure of carbon dioxide. PaO$_2$ = arterial partial pressure of oxygen. HCO$_3$ = serum bicarbonates. SaO$_2$ = arterial oxygen saturation of haemoglobin.

Pronation and lateral decubitus positions were started in 5 out of 10 patients undergoing h-CPAP in order to further improve ABG and clinical parameters, two of them eventually required ETI. Lateral decubitus was well tolerated by all patients, prone positioning was feasible and well tolerated in two cases, both at 22 weeks' gestation.

No adverse events, including pneumothorax, pneumomediastinum, hemodynamic instability, or venous thrombosis of the upper limbs, were observed during h-CPAP application.

We did not identify any difference in baseline demographic variables between patients requiring h-CPAP *versus* room air or oxygen only, Table 3. However, women who required h-CPAP application showed higher serum C reactive protein (CRP) and more extensive (≥3 lobes and bilateral) involvement at chest X-ray upon hospital admission.

Interestingly, assessment of temporal distribution of cases during the three pandemic waves showed substantially higher rates of h-CPAP requirement during the third wave.

## Maternal and perinatal outcomes

All patients, including those requiring h-CPAP and ETI with IMV, completely recovered from COVID-19. There were no cases requiring extracorporeal circulation oxygenation application due to persistent hypoxemic status after ETI with IMV.

Pregnancy is still ongoing in 7 (17.1%) women, 3 of whom required h-CPAP application during their hospital stay; in one case, ETI was performed at 22 weeks' gestation due to clinical

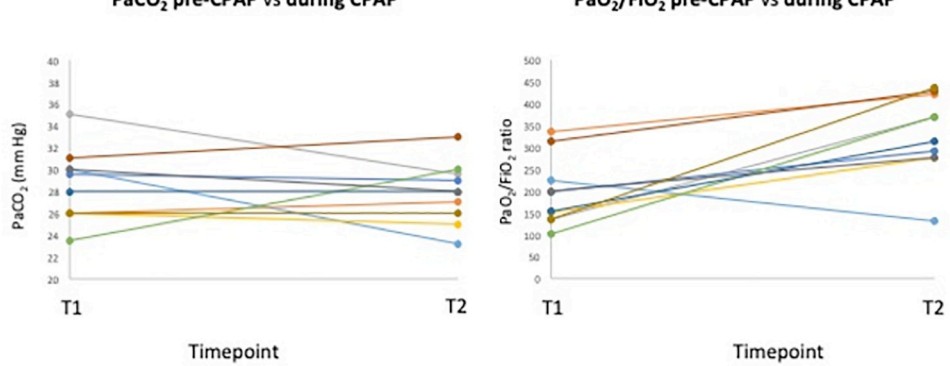

**Fig 3. Trajectories of PaO$_2$/FIO$_2$ and PaCO$_2$ between pre-CPAP (T1) and during CPAP application (T2).**
Footnotes: CPAP = continuous positive airway pressure. PaO$_2$/FIO$_2$ = arterial partial pressure of oxygen to fraction of inspired oxygen ratio. PaCO$_2$ = arterial partial pressure of carbon dioxide.

**Table 3. Demographic and clinical characteristics of study participants according to the need of h-CPAP.**

| Variables | Room air or oxygen only | h-CPAP application | p-value |
|---|---|---|---|
| | N = 31 | N = 10 | |
| **Timing of infection** | | | |
| 1st wave (Mar-Aug 2020) | 10 (32.2) | 0 | 0.040 |
| 2nd wave (Sept-Dec 2020) | 16 (51.6) | 3 (30.0) | 0.205 |
| 3rd wave (Jan-Mar 2021) | 5 (16.1) | 7 (70.0) | 0.003 |
| **Demographic and obstetric variables** | | | |
| Maternal age (years) | 31.6 ±4.9 | 33.3 ± 6.4 | 0.397 |
| BAME | 7 (22.6) | 3 (30.0) | 0.683 |
| Pregestational BMI $\geq$30 Kg/m$^2$ | 4 (12.9) | 4 (40.0) | 0.082 |
| Comorbidities | 6 (19.4) | 1 (10.0) | 0.660 |
| Nulliparity | 10 (32.3) | 4 (40.0) | 0.712 |
| Pregnancy complications | 15 (48.4) | 6 (60.0) | 0.719 |
| GA at hospital admission (weeks) | 31.4 ± 5.6 | 29.4 ± 5.0 | 0.318 |
| **Laboratory data at hospital admission** | | | |
| White blood cells (10^3/uL) | 7.93 ± 3.3 | 9.49 ± 4.3 | 0.260 |
| Lymphocytes (10^3/uL) | 1.2 ± 0.6 | 1.1 ± 0.7 | 0.560 |
| Platelets (10^3/uL) | 202.1 ± 68.5 | 253.8 ± 125.3 | 0.125 |
| D-dimer (ng/mL) | 713.4 ± 801.5 | 651.0 ± 539.5 | 0.858 |
| | (n = 28) | (n = 8) | |
| Fibrinogen (mg/dL) | 512.6 ±144.2 | 524.8 ± 157.4 | 0.839 |
| ALT (U/L) | 29.7 ± 34.8 | 38.9 ± 28.1 | 0.478 |
| LDH (U/L) | 185.3 ±39.3 | 213.0 ± 55.3 | 0.111 |
| CRP (mg/dL) | 3.0 ± 2.9 | 6.1 ± 4.4 | 0.021 |
| **Chest X-ray findings upon hospital admission** | | | |
| Involvement of $\geq$3 lobes | 3 (9.7) | 8 (80.0) | <0.001 |
| Bilateral involvement | 14 (45.2) | 10 (100.0) | 0.002 |
| Lobe involvement | | | |
| • Right Lower | 19 (61.3) | 10 (100.0) | 0.021 |
| • Left Lower | 13 (41.9) | 10 (100.0) | 0.002 |
| • Right middle | 7 (22.6) | 9 (90.0) | <0.001 |
| • Lingula | 8 (25.8) | 9 (90.0) | <0.001 |
| • Right Upper | 1 (3.2) | 3 (30.0) | 0.039 |
| • Left Upper | 5 (16.1) | 2 (20.0) | 1.000 |

Footnotes: Data shown as mean ± standard deviation or number (percentage).

BAME, Black, Asian, and minor ethnicity; BMI, Body Mass Index; h-CPAP, helmet continuous positive airway pressure; GA, gestational age.

and ABG deterioration notwithstanding h-CPAP. No maternal or fetal complications have been diagnosed in these women as of May 25th, 2021.

Among the remaining 34 patients, in 4 (11.8%) cases an emergent caesarean section was performed due to worsening of maternal status requiring ETI with IMV between 31st and 34th weeks' gestation. In only one case, there was also a concomitant deterioration of fetal condition, recognized by an abnormal biophysical profile and oligohydramnios. In turn, a term delivery occurred in the other thirty patients, 21 of whom had a vaginal delivery. All women and their neonates have been discharged in good conditions.

## Discussion

In our cohort of 41 consecutive pregnant women hospitalised with SARS-CoV-2 pneumonia, 10 (24.4%) developed a moderate-to-severe ARF and h-CPAP proved to be feasible, safe and well-tolerated in these cases. h-CPAP significantly improved oxygenation in the great majority of patients (8 out of 10) compared to oxygen therapy. The application of a h-CPAP trial allowed us to identify PEEP-responder patients that may benefit from continued h-CPAP application.

Prior studies have reported the efficacy of h-CPAP in improving oxygenation in patients with community acquired pneumonia and moderate hypoxemic ARF when compared to standard oxygen therapy [11]. A recent case report has showed the feasibility of h-CPAP application also in a pregnant woman with SARS-CoV-2 pneumonia and hypoxemic ARF, however no direct comparison between ABG values before and during h-CPAP application was available [9]. Therefore, our study is the first to describe the effect of h-CPAP on ABG values in pregnant women with SARS-CoV-2 pneumonia. The application of a h-CPAP trial allowed us to customize and maximize the effectiveness of therapy and to prevent inappropriate PEEP use, as already shown by *Paolini et al.* in a cohort of non-pregnant patients with pneumonia [12].

Despite an initial efficacy in improving gas exchanges, 3 out of 8 PEEP-responder patients faced a worsening of respiratory failure requiring ETI. Furthermore, both the PEEP non-responder women required ETI. A recent summary of available evidence reported only one case of worsening of respiratory failure requiring escalation to IMV out of 18 cases of pregnant women with COVID-19 requiring non-invasive ventilation [9]. Given the paucity of cases described so far, it is impossible to draw definitive conclusions on the efficacy of h-CPAP in preventing ETI in pregnant women and on the prognostic role of PEEP responsiveness. Nevertheless, continuous monitoring in an appropriate setting (e.g., high-dependency unit) during h-CPAP application should be mandatory to promptly identify early signs of ARF deterioration. Furthermore, lateral decubitus and prone positioning were feasible and well tolerated in 5 and 2 cases, respectively, in our cohort. However, no conclusions can be drawn on the efficacy of different positioning in improving gas exchange given the small number of cases and the heterogeneity of the decubitus.

Almost one out of four (24%) of the women hospitalised with SARS-CoV-2 pneumonia in our cohort showed a deterioration of clinical and ABG values which prompted h-CPAP use. This is higher than the 9.2% rate reported by a Chinese study on pregnant women hospitalised for SARS-CoV-2 pneumonia and requiring non-invasive ventilation during the first two months of the outbreak in China [15]. The evaluation of a longer time-frame of the pandemic, encompassing three subsequent waves of infection, as we did in our study, may explain this difference since we observed more severe cases requiring CPAP and ETI during the latest waves compared to the first one. In line with this observation, *Kadiwar et al.* has recently reported a substantial increase in pregnant women with severe COVID-19 during the second pandemic wave versus the first one [16]. Possible explanations of such finding have been speculated, including infection with more pathogenic variants of SARS-CoV-2 and increase in the total number of COVID-19 cases. Unfortunately, we could not investigate the presence of SARS-CoV-2 variants in our cohort due the impossibility to retrospectively process the nasopharyngeal swabs to search for alternative strains of SARS-CoV-2.

We identified a higher inflammatory response, evaluated through serum CRP levels, and a more extensive radiological lung involvement on admission in women requiring h-CPAP application. Of note, both conditions could be considered indices of disease severity [17, 18].

Iatrogenic preterm delivery by cesarean section due to worsening of maternal status and need of ETI with IMV was performed in four out of ten women with moderate-to-severe SARS-CoV-2-induced ARF requiring h-CPAP. Thus, the use of h-CPAP alone led to clinical improvement of the mother and allowed the safe continuation of pregnancy in 50% of the patients with hypoxemic ARF [7, 8, 15, 19, 20]. Also, a conservative management was successfully chosen for the mid-second trimester case (22 weeks' gestation) requiring ETI with IMV [21, 22]; this pregnancy is currently ongoing with no evidence of maternal or fetal complication as of May 25th, 2021.

Among the main strengths of our study, we acknowledge the inclusion of consecutive patients from a single tertiary care center with standardized protocols to manage COVID-19. This allowed to apply the same clinical protocol, including the h-CPAP trial, to all cases, although the monocentric design of the study limits its generalizability. Additional limitations are the limited sample size, which prevented us to accurately appreciate potential factors associated to the worsening of respiratory failure, and the impossibility to retrospectively evaluate the presence of SARS-CoV-2 variants in our cohort.

Future studies should aim to clarify the causes of the increased rate of moderate-to-severe ARF in pregnant women during the latest pandemic waves and to better evaluate the prognostic factors associated to ARF deterioration.

In conclusion, h-CPAP application proved to be feasible, safe and well-tolerated in pregnant women with moderate-to-severe hypoxemic ARF due to SARS-CoV-2 pneumonia in the setting of a high-dependency respiratory unit. h-CPAP was effective in improving oxygenation compared to oxygen therapy, but its role in preventing ETI still needs to be clarified. The application of a h-CPAP trial may allow a maximization of CPAP effectiveness, preventing the inappropriate PEEP use. Moderate-to-severe ARF was more frequently observed during the third pandemic wave (January-March 2021) and higher serum CRP levels as well as more extensive radiological lung involvement upon hospital admission were more frequently identified in patients requiring h-CPAP.

## Supporting information

**S1 Dataset.**
(XLSX)

## Author Contributions

**Conceptualization:** Paola Faverio, Sara Ornaghi, Anna Stainer, Patrizia Vergani, Alberto Pesci.

**Data curation:** Paola Faverio, Sara Ornaghi, Anna Stainer, Francesca Invernizzi, Mara Borelli, Federica Brunetti, Laura La Milia, Valentina Paolini, Roberto Rona.

**Formal analysis:** Paola Faverio, Sara Ornaghi.

**Investigation:** Paola Faverio, Sara Ornaghi.

**Methodology:** Paola Faverio, Sara Ornaghi.

**Supervision:** Alberto Pesci.

**Writing – original draft:** Paola Faverio, Sara Ornaghi.

**Writing – review & editing:** Paola Faverio, Sara Ornaghi, Anna Stainer, Francesca Invernizzi, Mara Borelli, Federica Brunetti, Laura La Milia, Valentina Paolini, Roberto Rona, Giuseppe Foti, Fabrizio Luppi, Patrizia Vergani, Alberto Pesci.

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
