## [Decision Letter · Decision Letter 0]

31 Aug 2021

PONE-D-21-18341

Feasibility of CPAP application and factors associated with worsening of respiratory failure in pregnant women with SARS-CoV-2 pneumonia: experience of a tertiary care centre

PLOS ONE

Dear Dr. Faverio,

Thank you for submitting your manuscript to PLOS ONE. After careful consideration, we feel that it has merit but does not fully meet PLOS ONE’s publication criteria as it currently stands. Therefore, we invite you to submit a revised version of the manuscript that addresses the points raised during the review process.

We look forward to receiving your revised manuscript.

Kind regards,

Tai-Heng Chen, M.D.

Academic Editor

PLOS ONE

Journal Requirements:

Reviewers' comments:

Reviewer's Responses to Questions

**Comments to the Author**

1. Is the manuscript technically sound, and do the data support the conclusions?

Reviewer #1: Partly

Reviewer #2: Yes

2. Has the statistical analysis been performed appropriately and rigorously? 

Reviewer #1: No

Reviewer #2: Yes

3. Have the authors made all data underlying the findings in their manuscript fully available?

Reviewer #1: Yes

Reviewer #2: Yes

4. Is the manuscript presented in an intelligible fashion and written in standard English?

Reviewer #1: Yes

Reviewer #2: Yes

5. Review Comments to the Author

Reviewer #1: Had been an honor to review an original article titled Feasibility of CPAP application and factors associated with worsening of respiratory failure in pregnant women with SARS-CoV-2 pneumonia: experience of a tertiary care centre”, for the initially congratulate to authors by their effort during the COVID-19 pandemic

About the article, I have an observation: the authors used the term ““ASSOCIATED FACTORS” however, the análisis plan performed don’t show any statistical association but means the difference between groups

I recommend performing a logistic regression (univariate and bivariate) or another analysis to justify the association. In my criteria, this would be valuable information to accept this article

Reviewer #2: This paper is very interesting and well written. It brings up clinical evolution among COVID-19 pregnant women, especially under the use of helmet CPAP. The study comprised 41 patients, which is a reasonable number in this context. Although retrospective, there is a systematic protocol to be followed in this center, which makes it possible to draw important conclusions.

The use of chest x-ray instead of computed tomography probably underestimated the extension of the disease, but I think this is acceptable due to the pregnancy.

Some points of concern:

1] Why should one expect that helmet CPAP wouldn’t be feasible to apply in a pregnant woman? Make a comment regarding this issue in the introduction.

2] The classic cutoffs to define acute respiratory failure are PaO2 of 60 mmHg and PaCO2 of 50 mmHg. Looking at Table 2, PaO2 interquartile range varied from 77.25 to 85.75 on admission. Patients got worse and probably showed lower values that did not appear at this table because they were on oxygen therapy. In the discussion section, the authors also mentioned moderate to severe ARF. What were the specific criteria used to define and categorize acute respiratory failure?

3] How was FIO2 calculated in patients under oxygen therapy by nasal catheter? There are some practical rules, but they could be inaccurate. I think the authors should describe how they did that.

4] FIO2 (capital letter I) should be written instead of FiO2, according to standardization of definitions and symbols in respiratory physiology (Fed Proc 1950; 9:602-5).

5] I think that “helmet CPAP” (I suggest h-CPAP or H-CPAP) is advisable whenever you need to refer to it, because “CPAP” is traditionally associated to a face mask.

6. PLOS authors have the option to publish the peer review history of their article (what does this mean?). If published, this will include your full peer review and any attached files.

Reviewer #1: No

Reviewer #2: No

---

## [Author Response · Author response to Decision Letter 0]

7 Sep 2021

Journal Requirements

We changed the files’ name as per Journal style.

2. We note that you have indicated that data from this study are available upon request. PLOS only allows data to be available upon request if there are legal or ethical restrictions on sharing data publicly.

Prompts

a. If there are ethical or legal restrictions on sharing a de-identified data set, please explain them in detail (e.g., data contain potentially sensitive information, data are owned by a third-party organization, etc.) and who has imposed them (e.g., an ethics committee). Please also provide contact information for a data access committee, ethics committee, or other institutional body to which data requests may be sent.

b. If there are no restrictions, please upload the minimal anonymized data set necessary to replicate your study findings as either Supporting Information files or to a stable, public repository and provide us with the relevant URLs, DOIs, or accession numbers.

We uploaded the minimal anonymized data set as requested.

The study was approved by the Institutional Review Board of the University of Milano Bicocca (3140/2020). Written consent was obtained by patients included in the study. We added this information in the text.

Review Comments to the Author

Reviewer #1: 

Had been an honor to review an original article titled Feasibility of CPAP application and factors associated with worsening of respiratory failure in pregnant women with SARS-CoV-2 pneumonia: experience of a tertiary care centre”, for the initially congratulate to authors by their effort during the COVID-19 pandemic 

We thank the Reviewer for his/her appreciation of our work, highlighting our research effort during the SARS-CoV-2 pandemic.

R1.C1

About the article, I have an observation: the authors used the term ““ASSOCIATED FACTORS” however, the análisis plan performed don’t show any statistical association but means the difference between groups

I recommend performing a logistic regression (univariate and bivariate) or another analysis to justify the association. In my criteria, this would be valuable information to accept this article

R1.R1

The Referee pointed out that the term ‘associated factors’ has been inaccurately used in the manuscript since the analysis we performed, i.e., univariate, cannot demonstrate an association (between the need of CPAP and patients’ characteristics) but only the presence of differences between the study groups (“room air” or “oxygen only” versus CPAP). Thus, the Referee suggested to perform a logistic regression analysis. 

We agree with the referee that the term ‘association’ has been used incorrectly. Since this is a retrospective cohort study, statistical analyses can only provide information regarding the presence or not of significant differences between the assessed groups. We have now addressed this point by substituting the term ‘association’ with more statistically appropriate terms throughout the manuscript.

As per performing additional analyses, a logistic regression analysis could be helpful if significant differences observed at the univariate analysis are considered to be biased by differences in patients’ characteristics. In our cohort, patients requiring CPAP had similar demographic and obstetric variables compared to patients in room air or requiring oxygen only, thus preventing an effect of confounding factors on the observed differences in CRP levels and rate of patients with >3 lobes involvement. In addition, the small size of the CPAP group (n=10) would substantially limit the validity of a logistic regression analysis. 

Reviewer #2: 

This paper is very interesting and well written. It brings up clinical evolution among COVID-19 pregnant women, especially under the use of helmet CPAP. The study comprised 41 patients, which is a reasonable number in this context. Although retrospective, there is a systematic protocol to be followed in this center, which makes it possible to draw important conclusions.

The use of chest x-ray instead of computed tomography probably underestimated the extension of the disease, but I think this is acceptable due to the pregnancy.

Some points of concern:

R2.C1

Why should one expect that helmet CPAP wouldn’t be feasible to apply in a pregnant woman? Make a comment regarding this issue in the introduction.

R2.R1

We thank the Reviewer for rising this point. In the available literature, evidence supports the feasibility of helmet CPAP application in adult patients with community acquired pneumonia (Cosentini et al. Chest. 2010;138:114-20). In contrast, no evidence is available regarding helmet CPAP utilization and feasibility in pregnant women with SARS-CoV-2 pneumonia. However, as indicated by the Reviewer, we explained this issue in the introduction of the manuscript.

R2.C2

The classic cutoffs to define acute respiratory failure are PaO2 of 60 mmHg and PaCO2 of 50 mmHg. Looking at Table 2, PaO2 interquartile range varied from 77.25 to 85.75 on admission. Patients got worse and probably showed lower values that did not appear at this table because they were on oxygen therapy. In the discussion section, the authors also mentioned moderate to severe ARF. What were the specific criteria used to define and categorize acute respiratory failure?

R2.R2

We thank the Reviewer for this comment. None of the patients required oxygen supplementation on hospital admission. While the mean values of PaO2 pre-CPAP (T1) and during CPAP (T2) indicated in Table 2 are obtained in patients receiving oxygen therapy, therefore, in order to evaluate respiratory failure severity, we utilized the PaO2 / FIO2 ratio. This solid instrument allowed us to define respiratory failure severity.

R2.C3

How was FIO2 calculated in patients under oxygen therapy by nasal catheter? There are some practical rules, but they could be inaccurate. I think the authors should describe how they did that.

R2.R3

All ABG analyses were performed in patients wearing a Venturi mask to obtain a precise FIO2 evaluation. However, during the hospital stay, Venturi mask was alternated to nasal catheter in patients with mild respiratory impairment to improve patients’ comfort.

R2.C4

FIO2 (capital letter I) should be written instead of FiO2, according to standardization of definitions and symbols in respiratory physiology (Fed Proc 1950; 9:602-5).

R2.R4

We thank the Reviewer for this observation. The term “FiO2” has now been substituted with the term “FIO2” throughout the manuscript. 

R2.C5

I think that “helmet CPAP” (I suggest h-CPAP or H-CPAP) is advisable whenever you need to refer to it, because “CPAP” is traditionally associated to a face mask.

R2.R5

We agree with the Reviewer’s comment and the abbreviation ‘h-CPAP’, as suggested, has now been used.

---

## [Decision Letter · Decision Letter 1]

5 Oct 2021

Feasibility of CPAP application and variables related to  worsening of respiratory failure in pregnant women with SARS-CoV-2 pneumonia: experience of a tertiary care centre

PONE-D-21-18341R1

Dear Dr. Faverio,

We’re pleased to inform you that your manuscript has been judged scientifically suitable for publication and will be formally accepted for publication once it meets all outstanding technical requirements.

Kind regards,

Alessandro Marchioni

Academic Editor

PLOS ONE

Additional Editor Comments (optional):

Dear Paola,

 Thank you for your submission. Safety and feasibility of the CPAP with helmet in pregnant women with SARS-CoV-2 pneumonia is a topic of extreme interest and can be of help to clinicians who are facing the ongoing pandemic.

Reviewers' comments:

Reviewer's Responses to Questions

**Comments to the Author**

1. If the authors have adequately addressed your comments raised in a previous round of review and you feel that this manuscript is now acceptable for publication, you may indicate that here to bypass the “Comments to the Author” section, enter your conflict of interest statement in the “Confidential to Editor” section, and submit your "Accept" recommendation.

Reviewer #1: All comments have been addressed

Reviewer #2: All comments have been addressed

2. Is the manuscript technically sound, and do the data support the conclusions?

Reviewer #1: Yes

Reviewer #2: Yes

3. Has the statistical analysis been performed appropriately and rigorously? 

Reviewer #1: Yes

Reviewer #2: Yes

4. Have the authors made all data underlying the findings in their manuscript fully available?

Reviewer #1: Yes

Reviewer #2: Yes

5. Is the manuscript presented in an intelligible fashion and written in standard English?

Reviewer #1: Yes

Reviewer #2: Yes

6. Review Comments to the Author

Reviewer #1: Dear authors

I am very delighted with the corrections performed by you to this paper. In the pandemic context and although is a retrospective study, yours had been a valuable effort to many healthcare workers in the front-line against COVID-19

Reviewer #2: (No Response)

7. PLOS authors have the option to publish the peer review history of their article (what does this mean?). If published, this will include your full peer review and any attached files.

Reviewer #1: No

Reviewer #2: No

---

## [Editor Report · Acceptance letter]

11 Oct 2021

PONE-D-21-18341R1 

Feasibility of CPAP application and variables related to worsening of respiratory failure in pregnant women with SARS-CoV-2 pneumonia: experience of a tertiary care centre 

Dear Dr. Faverio:

I'm pleased to inform you that your manuscript has been deemed suitable for publication in PLOS ONE. Congratulations! Your manuscript is now with our production department. 

Kind regards, 

on behalf of

Dr. Alessandro Marchioni 

Academic Editor

PLOS ONE